# Targeting Protein Degradation Pathways in Tumors: Focusing on their Role in Hematological Malignancies

**DOI:** 10.3390/cancers14153778

**Published:** 2022-08-03

**Authors:** Anna Wolska-Washer, Piotr Smolewski

**Affiliations:** Department of Experimental Hematology, Medical University of Lodz, 93-510 Lodz, Poland; adablju@tlen.pl

**Keywords:** unfolded protein response, endoplasmic reticulum stress, autophagy, chaperone-mediated autophagy, macroautophagy, heat shock proteins, PROTAC, LYTAC, hematological malignancies

## Abstract

**Simple Summary:**

Proteostasis is a balance between protein formation and degradation. All cells maintain their proteome throughout their lifespan. Malignant cells rely on protein degradation mechanisms to replenish cellular supplies of amino acids. Interference with the protein degradation machinery has become an effective method of fighting hematological malignancies, i.e., using proteasome inhibitors in cases of multiple myeloma and other lymphomas. Newer modalities have emerged, including proteolysis-targeting chimeras (PROTACs), lysosome-targeting chimeras (LYTAC), and other techniques for protein degradation that lead to cellular apoptosis. The current review focuses on targeted protein degradation and its use in the treatment of hematological malignancies.

**Abstract:**

Cells must maintain their proteome homeostasis by balancing protein synthesis and degradation. This is facilitated by evolutionarily-conserved processes, including the unfolded protein response and the proteasome-based system of protein clearance, autophagy, and chaperone-mediated autophagy. In some hematological malignancies, including acute myeloid leukemia, misfolding or aggregation of the wild-type p53 tumor-suppressor renders cells unable to undergo apoptosis, even with an intact p53 DNA sequence. Moreover, blocking the proteasome pathway triggers lymphoma cell apoptosis. Extensive studies have led to the development of proteasome inhibitors, which have advanced into drugs (such as bortezomib) used in the treatment of certain hematological tumors, including multiple myeloma. New therapeutic options have been studied making use of the so-called proteolysis-targeting chimeras (PROTACs), that bind desired proteins with a linker that connects them to an E3 ubiquitin ligase, resulting in proteasomal-targeted degradation. This review examines the mechanisms of protein degradation in the cells of the hematopoietic system, explains the role of dysfunctional protein degradation in the pathogenesis of hematological malignancies, and discusses the current and future advances of therapies targeting these pathways, based on an extensive search of the articles and conference proceedings from 2005 to April 2022.

## 1. Introduction

Maintaining a stable proteome is essential for the survival of every cell. While precisely regulated protein synthesis is vital, it is equally important that unwanted or abnormal proteins are degraded via various cellular processes. Protein degradation involves the unfolded protein response (UPR) with proteasome-based protein clearance, and autophagy (macrophagy, microautophagy, and chaperone-mediated autophagy). Protein homeostasis, also referred to as proteostasis, is a continuous process of maintaining the integrity of cellular proteins in higher organisms [1]. Proteins are continuously produced and degraded at a level required to maintain proteostasis. The levels of such control are determined by each given cell (autonomous), and might also be orchestrated by the nervous system (non-autonomous) to provide more centralized supervision [2]. 

Protein interactions, protein trafficking to subcellular compartments, and finally degradation, are regulated by a net of interactions between distinct pathways, and any dysregulation can result in abnormal protein folding. Many diseases that are accompanied by abnormal protein accumulation are caused by either the overproduction of normal proteins that misfold eventually (i.e., secretory plasma cells in multiple myeloma), overproduction of misfolded proteins that overcome the natural buffering mechanisms, or a defect in the control pathways that are responsible for halting protein genesis and starting the degradation of abnormal proteins (aging, amyloidosis) [3,4]. In addition, cellular protein control processes decline with age, leading to aging, degenerative diseases, or cancer. 

Proteins are susceptible to thermal damage because of the necessity of keeping their structures flexible for conformational changes depending on the required circumstances i.e., protein–protein interactions. Therefore, the protein degradation systems in malignant cells have become an important topic for study [5]. Cellular sensors of stress include proteins, RNA, DNA, and membrane lipids [6]. At high-stress levels, control is provided by the unfolded-protein response (UPR) and autophagy. 

The very first step in the UPR is the intramembrane proteolysis-activated ATF6, which translocates to the Golgi body. There, it undergoes cleavage resulting in the formation of ATF6p50, which travels into the nucleus to induce several genes, improving the ER protein quality control and secretory function, as well as increasing its size [7,8,9]. The second UPR pathway—IRE1α—promotes the activation of a potent transcription factor X-box binding protein 1 (XBP1) by processing its encoding mRNA [10]. Activated XBP1 (XBP1s) is responsible at this stage for the protein trafficking, folding, and degradation that occurs via the ER-associated degradation (ERAD), which targets ER-localized proteins within the proteasome in the cytosol [11].

Finally, the third ER-related UPR is mediated by the PKR-like ER kinase PERK (also called eukaryotic translation initiation factor 2 alpha kinase 3 EIF2AK3), which undergoes dimerization and phosphorylation [11]. The active PERK phosphorylates eukaryotic initiation factor 2aplha (eIF2alpha), which remains inactive and results in reduced general protein translation and flux into the ER lumen. Moreover, some mRNAs are permitted to proceed to translation. Among these is activating transcription factor 4 (ATF4), which induces either pro-survival genes (antioxidant response, amino acid synthesis, autophagy) or pro-apoptotic genes, depending on the cell type, and duration of stress [12]. The activated pro-survival genes include a cellular inhibitor of apoptosis (cIAP), myeloid cell leukemia 1 (MCL1), nuclear factor kappa B (NF-kB), and nuclear protein transcription regulator 1 (Nupr1) [12,13,14,15]. One of the major genes that initiate apoptosis is C/EBP homologous protein (CHOP, or GADD153) [16,17]. It is a transcription factor normally present at low levels, activation of which leads to apoptosis mediated by downregulation of Bcl2 and depletion of glutathione; this renders the cell prone to ROS under stressful conditions [17]. All three ER stress pathways activate autophagy, as reflected by the transcription of multiple autophagy-related genes (ATG) [18]. Another link between ER stress and protein degradation is through ER-associated degradation (ERAD) and the ubiquitin-proteasome system (UPS) [19]. The main protein degradation pathways include the ubiquitin-proteasome system (UPS) and the autophagy-lysosome pathway [20,21]. These can both be modified by chaperone molecules.

## 2. Ubiquitin-Proteasome System

Interaction of the ubiquitin-activating (E1), -conjugating (E2), and -ligase (E3) enzymes results in the selective ubiquitylation of the substrate at certain lysine residues—either mono-, multi- or poli-ubiquitylation (poly-Ub) [22] (Figure 1). The ubiquitylated lysine residue determines the fate of the protein and therefore ubiquitylation can regulate cell division, mitophagy, endoplasmic reticulum-associated degradation (ERAD), lysosomal degradation, kinase modification, inflammation, DNA repair, endocytosis, transcriptional regulation, and proteasomal degradation. Another set of enzymes, deubiquitinases (DUBs), removes ubiquitin from its poly-Ub chains and produces free ubiquitin (Ub) by modifying its precursors.

Another step in protein degradation by UPS involves a multi-subunit ATP-dependent protease called 26S proteasome [23]. This is probably the most important protease in the cell, and degrades proteins in the cytosol and the nucleus; briefly, it removes poly-Ub, unfolds the protein, and translocates it into the inner chamber of the complex. Finally, the outcome of the proteasome activity produces peptides approximately 10 amino acids long. A 26S proteasome is structured as a chamber (the 20S core protease) blocked at one or both ends by a 19S regulatory particle [24]. There are two main subtypes of proteasomes—the constitutive proteasome and the immunoproteasome [25]. The expression of the constitutive proteasome is universal across all cell types, while immunoproteasomes are expressed in cells of organs involved in the immune system, including the spleen, thymus, lung, liver, kidney, colon, small intestine, and antigen-presenting cells. Their expression can be induced by interferon gamma to enhance MHC class I antigen presentation in non-immune cells [24].

Synthesis of the 26S proteasome requires a significant amount of energy, and there are distinct mechanisms that regulate adequate production of the proteasome elements. It has been found that there are virtually no excessive subunits, but a stoichiometric amount of each component [26]. Any overproduced elements are either sequestered or degraded, allowing for tight regulation. The expression of the proteasome subunits is under the control of several transcription factors—nuclear transcription factor Y (NF-Y), Forkhead box protein O4 (FOXO4), signal transducer and activator of transcription 3 (STAT3), nuclear factor erythroid 2-related factor 1 (NRF1), and NRF2 [24]. It has been observed that proteasome inhibition leads to the increased production of proteasome subunits, reflected by an increase in the expression of all 26S proteasome subunit mRNAs [27]. These quantitative changes are also accompanied by an increase in proteasomal caspase-like, trypsin-like, and chymotrypsin-like peptidase quality, and allow Burkitt lymphoma cells to escape the proteasome inhibition [28].

Proteasome production and function are also influenced by post-translational modifications, including acetylation, ADP-ribosylation, glycosylation, methylation, myristoylation, oxidation, phosphorylation, SUMOylation, ubiquitylation, and proteolytic processing [24,29]. Protein kinase A (PKA) and dual specificity tyrosine-phosphorylation-regulated kinase 2 (DYRK2) are each kinases that augment the activity of the 26S proteasome. DYRK2 was found to induce degradation of tumor-suppressor proteins and thus promote cancer growth, whereas its lower expression in some tumors was associated with poorer outcomes [30]. On the other hand, apoptosis signal-regulating kinase 1 (ASK1), which is activated by oxidative stress, promotes apoptosis by inhibiting proteasome function [31]. Therefore, there may be an opportunity for phosphorylation-modifying drugs to change the activity of the 26S proteasome in neurodegenerative or malignant diseases, but further research is necessary as some modifications may initiate dual responses.

During active protein synthesis, proteasomes are actively relocated into the nucleus and exported back into the cytosol at quiescent periods of the cell cycle. In mammalian neurons, plasma membrane-bound proteasome subunits were found to regulate peptides that were formed upon signal transduction [32].

Finally, proteasomes can be regulated by degradation. For example, upon the induction of apoptosis, the terminal regulatory particle (RP) subunits are cleaved by caspase-3, which leads to the destruction of the active proteasome [33]. A different mechanism is offered by either selective or non-selective autophagy, based on the trigger. During nutrient deprivation, random bulk autophagy, including proteasome autophagy, takes place to restore cellular stocks. Selective autophagy of proteasomes can be induced by the inhibition of proteasomes and mediated by tagging through the ubiquitination of proteasomes. Another mechanism relies on the autophagy receptor p62, which selects proteasomes for starvation-induced autophagosomal degradation, while other organelles are taken up randomly [33,34].

The UPS pathway is involved in every aspect of cell life, with ligases that can target a multitude of peptides and proteasomes capable of destroying them. The modulating potential of the UPS is thus underpinned by its control over numerous cellular processes. It has also been found to be involved in the development of several diseases characterized by neurodegeneration or immune dysfunction, as well as malignancies and aging [21]. Control of the proteasome is also necessary to prevent unwanted degradation of essential proteins. Deubiquitylation is the process of recycling ubiquitin and is a checkpoint on the proteasome [35]. Approximately 100 DUBs can be categorized among at least seven classes: the ubiquitin-specific proteases (USPs), the ubiquitin C-terminal hydrolases (UCHs), the ovarian tumor proteases (OTPs), the Machado-Josephin domain proteases (MJDs), the JAB1/MPN+/MOV34 (JAMM) domain proteases, the monocyte chemotactic protein-induced proteins (MCPIPs), and the motif interacting with ubiquitin-containing DUB family (MINDY) [22]. Their role is to remove the ubiquitin chains from proteins and prevent their degradation by the proteasome. The inappropriate activity of the whole system of ubiquitylation and deubiquitylation might directly or indirectly affect various signaling pathways and result in the development of cancer.

## 3. Autophagy

Autophagy is another complex mechanism of protein degradation that complements the UPS (Figure 2). The process involves a sequence of steps: induction, nucleation, elongation with substrate isolation, and fusion with a lysosome [36]. Stress activates the Unc-51-like kinase 1 (ULK1), ATG13, ATG101, and FIP200 complexes that initiate the formation of an intracellular membrane. That reaction is augmented by PERK and IRE through 5’ adenosine monophosphate-activated protein kinase (AMPK), but negatively regulated by the mammalian target of rapamycin complex 1 (mTORC1), which is a receptor of energy and nutrient levels in the cell. The ULK1 complex phosphorylates Beclin-1 (BECN1), which in turn forms another complex including PIK3R4, BECN1, ATG14, and VPS34, responsible for creating autophagy vesicles. The latter complex also interacts with the antiapoptotic Bcl2, which works in concert with BECN1. The elongation of the vesicles is mediated by the active ligase-like complex ATG12-ATG5-ATG16L1 and LC3-PE. Finally, LC3-PE entraps the substrate in the cytoplasm to create an autophagosome, which subsequently fuses with the lysosomes. The cargo receptors responsible for connections between LC3 and the ubiquitin chains are p62, NBR1, Optineurin (OPTN), Toll Interacting Protein (TOLLIP), and TAXBP1. Fine tuning of autophagy is achieved by the interactions of a family of heat shock proteins—Hsc70, Hsp40, Hsp90—with KFERQ-like motifs in the target proteins. The motifs are characterized by a certain sequence of amino acids including hydrophobic, positively-, and negatively-charged residues with single glutamine on the C- and N-terminus [37].

Hematopoiesis is a continuous life-long process of generating various specialized cells from multipotent hematopoietic stem cells (HSCs). The most immature cells of hematopoiesis demonstrate an increased autophagy gene signature that preserves them from accumulating damage during starvation or in the absence of growth factors [38]. The inability of HSCs to alleviate cellular damage due to the loss of one of the ATGs (*Atg7*) was found to result in a build-up of mitochondria and ROS [39]. The observed anomalies led to increased proliferation and DNA damage, and subsequent leukemic transformation. Moreover, the abnormal HSCs were unable to reconstitute sound hematopoiesis in irradiated mice. Autophagy was found to be responsible for the loss of PML-RARα, as well as normal RARα, induced by all-trans retinoic acid (ATRA) and mediated by the UPS [40]. A similar mode of action is exerted by arsenic trioxide (As_2_O_3_) with covalent modification of PML-RARα and its sumoylation. Both compounds can induce degradation by autophagy of PML-RARα via the mTOR pathway, and that process is responsible for its basal turnover [41]. During normal hematopoiesis, and in malignancies, the UPS plays a crucial role in the regulation of e.g., c-kit, cyclin D, Bcl-2, and others [42]. Moreover, cell signaling might be modified by proteasomal degradation of key cytokine receptors after their activation. CBL and SOCS are examples of proteins that catalyze ubiquitylation of receptors including colony stimulation factor-1 (CSF1R), and platelet-derived growth factor receptor (PDGFR) [42]. Erythropoiesis represents another example of efficient autophagy in order to produce functional erythrocytes [43]. Experiments with targeted gene deletion resulted in impaired mitophagy and defective ribosomal clearance. Furthermore, molecular chaperones such as Hsp70 protect erythroblasts from excessive apoptosis under erythropoietin-induced physiologic mitochondrial depolarization and the release of apoptosis-inducing factors [44]. Thalassemia is one potential example of the deleterious effect of the accumulation of misfolded proteins. Free alpha-globin is structurally unstable and readily self-associates into tetramers. It has a dedicated erythroid-specific molecular chaperone called alpha-hemoglobin-stabilizing protein (AHSP), which keeps its native structure and prevents ROS overproduction [44]. Finally, T lymphocyte development also relies on intact autophagy and energy metabolism [45]. Deletion of Atg5 or Atg7 at early CD4- and CD8-double negative stages of T cell maturation led to increased apoptosis of thymocytes and mature peripheral lymphocytes. The formation and maintenance of memory T cells depend on autophagy [46,47]. Memory T lymphocytes demonstrate activated AMPK and diminished mTOR signaling, stimulating autophagy and allowing for the mitochondrial metabolism required to maintain quiescence rather than stemness. B lymphocytes present a similar dependency on autophagy during development through their early stages up to pre-B cells [45]. Peripheral B lymphocytes seem unaffected by impaired autophagy [48]. On the contrary, plasma cells have high levels of autophagy compared to B cells, as increased production of immunoglobulins poses metabolic stress on plasma cells [49]. Deletion of Atg5 in mice led to increased expression of the ER stress genes Xbp-1 and Bip, which resulted in enhanced antibody production. However, increased metabolic stress induced by immunoglobulin synthesis alongside high mitochondrial ROS production eventually leads to plasma cell death [49]. Autophagy is also essential in the maturation of monocytes into macrophages, and opposes caspase-mediated apoptosis [45]. Moreover, under pro-inflammatory conditions in the tumor microenvironment, increased autophagy leads to polarization towards tumor-associated macrophages (TAMs) [50].

There are several ways of targeting protein degradation pathways in various states, including hematological malignancies, and potentially active tools for this process have been proposed. Since clinical experience regarding those disorders remains limited, this review also mentions ongoing trials including some relating to solid malignant neoplasms.

## 4. Proteolysis-Targeting Chimeras (PROTACs) and SNIPERs

Numerous proteins have been described as drivers of tumorigenesis. However, many of these are not ideal targets for new drugs as they lack the classical pockets for ligands in their structure. Moreover, some, for example RAS oncoproteins, utilize a vast spectrum of substrates and are difficult to modify [51]; as the network of possible interactions and substrates is so great, they are generally believed to be untreatable by conventional approaches. The discovery of so-called molecular glues was made in the 1990s, when studies of ciclosporin A and tacrolimus and their mechanisms of action [52] indicated that certain molecules might bring distinct proteins into proximity (Figure 1). The rapid development of synthetic molecular glues followed these initial studies and offered new options for the modulation of protein–protein interactions, including tagging proteins for degradation; for a review see [53]. Proteolysis-targeting chimeras (PROTACs) emerged as a novel method for targeting proteins of interest (POIs) for subsequent degradation [54,55]. These are small bifunctional molecules that better enable the natural cellular ATP-dependent ubiquitin proteasome system (UPS) to break down the target proteins. These molecules consist of a ligand or a warhead that targets the POI, a distinct ligand that recruits an E3 ligase, and a linker that connects the ligands.

There are about 600 E3 ligases, in three major families: RING/U-box, HECT, and RING-Between-RING (RBR) [56,57,58,59]. However, most PROTACs utilize only CRBN or VHL E3 ligases, which may be disadvantageous due to E3 ligase mutations rendering them unable to engage in PROTAC activity [60]. Moreover, the expression of E3 ligases is tissue-specific and even varies between different cell metabolic stages. Tailoring PROTACs to a specific E3 ligase profile in a given cell target may perhaps facilitate their efficacy. Alternative E3 ligases with known ligands include mouse double minute 2 homolog (MDM2), cellular inhibitor of apoptosis protein-1 (cIAP1), X-linked inhibitor of apoptosis protein (XIAP), aryl hydrocarbon receptor (AhR), Kelch-like ECH-associated protein 1(KEAP1), and DDB1 and CUL4 associated factor 15 (DCAF15) [60]. The recognition of the substrate is guided by specific motifs called degrons, i.e., a free N-terminal amino acid (N-degron), C-degron, or hydroxylated proline residue in the Hypoxia-Inducible Factor 1 α (HIF1α). PROTACs drive POIs into the UPS degrading system by providing chemically-induced proximity of the ubiquitin substrates. The PROTAC itself remains intact and searches for another POI. The occupancy of the target or the affinity of the PROTAC play secondary roles, as the mode of action depends on the single contact that marks the POI for degradation. This so-called event-driven mechanism of action does not follow the conventional approach to drug design, and resembles a catalytic reaction under sub-stoichiometric conditions. Therefore, PROTACs show longer activity with less toxicity, as they require less total drug exposure. Thus, PROTACs have opened a new era of targeted agents that aim at undruggable proteins.

An interesting discovery was made in 2010, with cereblon (CRBN) being revealed as a target of immunomodulatory drugs (IMiDs) and a part of the DDB1-CRBN E3 ligase complex [61,62]. IMiDs block the ubiquitin-induced degradation of the endogenous substrate MEIS2 while recruiting new proteins: lymphoid transcription factors Ikaros and Aiolos (IKZF1 and IKZF3) and casein kinase 1α (CK1α). The latter are responsible for the therapeutic effect of IMIDs in multiple myeloma and del(5q) myelodysplastic syndrome. CRBN ligands have been identified as being among the most promising E3 ligase ligands, because of their strong binding to E3 ligases, well-recognized binding characteristics, and other physicochemical properties [63]. The science of PROTACs finally reached clinical trials in 2020 with two degraders of the androgen receptor (ARV-110, CC-94676) and one of the estrogen receptor (ARV-471) [64].

Table 1 The currently studied PROTACs in clinical trials.

### 4.1. BTK Pathway—Targeting PROTACs

A novel approach with targeting proteins for degradation was applied for Bruton tyrosine kinase (BTK)-driven hematological malignancies. BTK is a vital part of the signaling pathway downstream from the B-cell receptor (BCR), and its continuous activation is a hallmark of several hematological malignancies [65,66,67]. Inhibitors of BTK became a promising therapeutic modality in the treatment of e.g., chronic lymphocytic leukemia (CLL), Waldenstrom macroglobulinemia (WM), and mantle cell lymphoma (MCL) [65,68,69,70]. However, mutations in the BTK can render malignant cells refractory to its inhibition [71]. One method of overcoming inevitable resistance is using non-covalent BTKs such as pirtobrutinib (LOXO-305), nemtabrutinib (ARQ 531), fenebrutinib (GDC-0853), which do not permanently occupy the binding groove [71]. However, the requirement for a molecule to bind to the target in a specific way might still pose a risk of developing resistance by changing the conformation of the slot. Therefore, a non-classical approach that utilizes sheer contact with the BTK, regardless of the mutational status, became a plausible option in resistant cases.

Designing a functional PROTAC against BTK requires choosing an appropriate warhead with good binding properties in wild-type as well as mutated BTK, an E3 ligase ligand expressed in the targeted tissue/cell, and a linker to provide proximity for the ligase and the substrate. An elegant study was conducted in vitro by Zorba et al., which demonstrated the relations between various compounds to ensure functionality [72]. The warhead was a non-covalent reversible derivative of PF-06250112 BTK inhibitor that earlier proved successful in alleviating a mouse model of lupus [73]. The chosen E3 ligase was CRBN and the corresponding ligand used was pomalidomide. The study attempted to determine the perfect length of the pegylated linker from among 11 candidates. An exponential decrease in BTK levels started at 1 h of incubation but plateaued after 24 h. Moreover, the dose-dependence curve had a U-shape which was explained by the formation of binary complexes (BTK-PROTAC and PROTAC-CRBN) that competed with a functional ternary complex (BTK-PROTAC-CRBN)—the prozone or ‘hook’ effect [74]. Moreover, rapid recovery of BTK levels was observed upon washout within 24 h. Longer linkers (≥14 atoms) conferred potent degradation, compared to shorter linkers (≤9 atoms), and intermediate ones gave intermediate results.

This finding is supported by another study describing a shorter linker connected to a BTK-binding scaffold, making the whole structure longer and allowing for energetically preferred complexes [75]. The choice of E3 ligase also made a difference; despite similar cellular localization, VHL and IAP ligases were ineffective [72]. However, computational modeling predicted engagement of BTK and VHL in an energetically favorable ternary complex. The authors explained that the CRBN-associated E3 ligase may have greater flexibility and offer access to E2 active sites for a wider range of surface lysine. The studied PROTACs were also able to degrade lymphoid transcription factors ZFP91, IKZF1, and IKZF3, but not IKZF2, which confirmed the immunomodulatory effects of BTK PROTACs with a CRBN E3 ligand. Surprisingly, tissue/cell specificity was observed with reduced levels of BTK in the spleen but not in the lung in rats, regardless of the dose. The latter finding has no explanation at the moment, given the fact that both sites achieved similar PROTAC exposure. Perhaps some differences occur in the expression of the target, E3, or the deubiquitinase activity, among other possibilities.

A very promising approach against wild-type as well as C481S mutated BTK, was studied in vitro, using an ibrutinib derivative which lacked the propensity to bind BTK covalently at cysteine 481, linked to pomalidomide [76]. The final compound MT-802 with an eight-atom linker displayed the highest BTK degradation capacity in vitro. A 12-atom linker was also observed to have good potency, suggesting that this construct produced an efficient BTK-CRBN ternary complex. The researchers also noted that utilizing VHL produced modest target degradation only at higher micromolar concentrations (DC_50_ at 1.0–2.5 µM), compared to nanomolar concentrations required for the CRBN-based PROTAC MT-802 (DC_50_ at 9.1 nM, 50 min). The maximal BTK degradation with MT-802 was observed at 250 nM as early as 4 h, and a further increase of its concentration up to 2.5 µM did not induce any rebound in BTK concentration (the so-called ‘hook’ effect). The latter might be explained by positive cooperativity between the POI and the E3 ligase, and thus reduced formation of unproductive binary complexes [77]. MT-802 showed reduced or no binding of ITK, MKK7, and JAK3 kinases, among others, in contrast to ibrutinib. Moreover, MT-802 demonstrated similar activity to the ibrutinib binding of ERBB3 kinase without its degradation in the OVCAR8 cell line. Also, MT-802 did not degrade IKZF1 or IKZF3 transcription factors targeted by pomalidomide and CRBN. Degradation of the C481S BTK mutant was retained by MT-802 due to its event-driven mode of action, whereas ibrutinib showed a 40-fold decrease in its inhibitory feature. The effect of MT-802 was also demonstrated in primary CLL cell samples with a confirmed C481S BTK mutation from patients before and after ibrutinib relapse. MT-802 showed significant degradation of BTK in the wild-type and mutated BTK patient samples. Furthermore, MT-802 was also effective in reducing active Y223 phosphorylated BTK after relapse, which demonstrates additional features related to the event-driven strategy. The authors concluded that a successful PROTAC structure might be based on suboptimal binders (here the ibrutinib scaffold) due to the event-driven mode of action.

Another oral compound, NX-2127, was studied against a mutated form of Bruton tyrosine kinase (BTK) [78]. NX-2127 consists of a warhead that binds BTK, a linker, and a CRBN ligand. It has demonstrated activity at nanomolar concentrations in BTK wild-type and BTK mutated cells across human malignant cell lines and in animal models. Moreover, NX-2127 exerts IMiD-like action by degradation of IKZF1 and IKZF3 through CRBN. The latter effect is coupled with the stimulation of T lymphocytes with increased Il-2 production. This dual action warrants its further development for addressing human B-cell malignancies, especially those with BTK mutations rendering them resistant to covalent BTK inhibitors.

Sun et al. searched for a BTK-degrading PROTAC by studying a variety of combinations with ibrutinib and spebrutinib (CC-292, covalent BTKi) as the BTK-binding ligands, and pomalidomide and RG-7112 (MDM2 ligand) as E3 ligase ligands [79]. The combination of ibrutinib and the pomalidomide P13I exhibited the best degradation ability. P13I was then tested in the human ABC-DLBCL, HBL-1 cell line. The half-lives of wild-type and C481S BTK upon incubation with P13I were 4 and 3 h, respectively, at a concentration of 6.3 nM required to achieve DC_50_ (50% protein degradation concentration). Similar efficacy was demonstrated in the Burkitt’s lymphoma cell line (RAMOS), MCL cell line (Mino cells), and MM cell line, with DC_50_ of 8.5 nM, 9.2 nM, and 11.4 nM, respectively. The in vitro experiments with HBL-1 cells demonstrated a GI_50_ (50% growth inhibition concentration) for P13I of 1.5 nM, compared to 2.5 nM for ibrutinib. However, the IC_50_ (50% inhibitory concentration) against BTK for P13I was 95 nM, compared to 0.5 nM for ibrutinib, which proves that the P13I PROTAC acts through BTK degradation rather than inhibition. There was no effect on RAMOS cells, which do not rely on BTK signaling, thus confirming that P13I does not exert any non-specific cytotoxicity. P13I was also effective in inhibiting the C481S BTK mutated DLBCL cell line at 28 nM, compared to ibrutinib which showed no efficacy (700 nM). There was no off-target degradation by P13I of other kinases such as EGFR, ITK, or TEC even at 5 µM. These promising results led to a subsequent study with a new BTK degrader—L18I—based on lenalidomide as a CRBN partner [80]. L18I efficiently degraded a C481S BTK-harboring human ABC-DLBCL cell line, HBL-1, at a DC_50_ of 29 nM. The mutated C481S BTK half-life was < 2 h after incubation with L18I. In addition to C481S mutation, L18I was efficient at approximately 30 nM against other substitutions at the C481 position: C481T, C481G, and C481A. The 50% inhibition of cell proliferation concentration (GI_50_) was 64 nM in HBL-1 BTK cells expressing C481S BTK.

The downstream BCR molecules were efficiently inhibited by L18I, i.e., the phosphorylation of PLCγ-2, ERK1/2, and p38, which could not be achieved with ibrutinib. L18I showed remarkable efficacy in an MCL cell line expressing C481S BTK at concentrations as low as 10 nM. Further experiments involved xenograft murine models with C481S BTK HBL-1 cells. With intraperitoneal daily injections for two weeks, tumor regression was observed with 30 or 100 mg/kg. The same dosages of ibrutinib were used for comparison. L18I induced significant 36% and 63% reductions in tumor size with the given doses, respectively. Ibrutinib did not produce any tumor shrinkage. The safety evaluation showed increased body mass loss in mice treated with ibrutinib. Moreover, acute toxicity experiments showed good tolerability of L18I up to 300 mg/kg three weeks after the treatment. To assess the efficacy of combination therapy aiming at BTK and other BCR signaling molecules, C481S BTK HBL-1 cells were incubated with L18I and an SYK inhibitor, GS-9973. The combination resulted in a synergistic effect with an L18I GI_50_ concentration of 11 nM. Similarly, combination treatment with a PI3K inhibitor, copanlisib, resulted in even better L18I activity, with a GI_50_ of 5 nM.

A study by Xue et al. used BTK covalent inhibitor-based PROTACs in the K562 cell line of human chronic myelogenous leukemia [81]. Ibrutinib and PLS-123 (2,5-diaminopyrimidine-based compound) were modified and linked to pomalidomide or VH033 for CRBN and VHL, respectively. Several distinct linkers were used, and a reduction of BTK protein level by 50% was observed with concentrations of ibrutinib-based compounds ranging from 100 nM to 3 µM, while ibrutinib alone showed no decrease. PLS-123 was also effective compared with ibrutinib-based PROTAC at concentrations less than 300 nM. A substitution of pomalidomide with a VHL-recruiting VH032 showed a promising DC_50_ at 136 nM. Several other PROTACs were constructed and tested for BLK degradation in RAMOS cells. The most efficient results were obtained at 220 nM, with a maximum reduction of 75%. The authors concluded that covalent BTK inhibitors could be successfully incorporated into PROTACs with adjustments in the linker and E3 ligase to obtain excellent degraders for clinical use.

### 4.2. Bcl-xL—Targeting PROTACs

Another PROTAC of interest in hematology is DT2216, which targets Bcl-xL for degradation in T-cell lymphomas that greatly depend on the overexpressed proteins of the Bcl-2 family, such as Bcl-2, Bcl-xL, and Mcl-1 [82]. Bcl-xL is expressed in a subset of leukemia/lymphoma cells (AML, ALL, Hodgkin’s lymphoma, Burkitt’s lymphoma, DLBCL, FL, MZL, MM) and confers them resistant to chemotherapy [83]. The direct Bcl-xL inhibitor navitoclax (ABT263) induced dose-limiting thrombocytopenia and a reduced capacity of platelet adhesion through glycoproteins [84]. DT2216 delivers selective proteasomal degradation of Bcl-xL through the von Hippel-Lindau (VHL) E3 ligase without a detrimental effect on the platelets, because platelets do not express a significant amount of VHL [82]. The compound derived from the ABT263 Bcl-xL-binding moiety with a VHL ligand induced apoptosis in the MOLT-4 T-cell acute lymphoblastic leukemia cell line, with minimal activity on platelets. Weekly parenteral dosing of DT2216 almost completely blocked MOLT-R T-ALL xenograft growth in mice, without significant platelet reduction or a subsequent rebound effect of thrombocythemia (as seen with oral agent ABT263 in the same experiment). DT2216 was also effective in ABT263 refractory cases. It was suggested that combined inhibition of other BCL-2 family proteins might improve the proapoptotic impact. Inhibition by DT2216 combined with either a BCL-2 inhibitor (ABT263) or an MCL-1 inhibitor (S63845) resulted in the synergistic killing of human small cell lung cancer and multiple myeloma cell lines [82]. Increased chemosensitivity was observed upon combined treatment with DT2216 and conventional chemotherapy in triple-negative MDA-MB-231 breast cancer, PC-3 prostate, HepG2 liver, and SW620 colon cell lines. Moreover, the effect was also present in a patient-derived xenograft tumor model of resistant T-cell ALL, when DT2216 was combined with vincristine, dexamethasone, and L-asparaginase. The median overall survival of mice reached 72 days with combination treatment versus 55 days with DT2216 monotherapy, and 47 days with chemotherapy alone. Similarly, good results were seen in xenograft mouse models with D115 and 332X-luci T-ALL cells. DT2216 showed remarkable effects in another xenograft mouse model of human T-cell lymphoma (MyLa, MJ, MAC2A, and L82 cell lines) and T-cell prolymphocytic leukemia (T-PLL), with reduced platelet toxicity compared with ABT263 [85]. The addition of ABT199 (venetoclax) to DT2216 acted synergistically on T-cell lymphoma in vitro and was found more effective against T-PLL in xenograft models. The latter effect was paralleled by a significant reduction in Bcl-xL expression in T-PLL cells in the spleen. Venetoclax monotherapy induced increased Mcl-1 expression, which was blocked by DT2216. The abovementioned studies indicate that the dose-limiting normal tissue toxicity, encountered with a conventional direct inhibitor, could be avoided by using a PROTAC molecule targeting the POI with an E3 ligase that is not expressed in normal tissue. A tissue-specific E3 ligase ensures tissue-specific degradation of POI, with a warhead that might be too toxic when applied as a classical inhibitor. Moreover, combined treatment with either conventional inhibitors or chemotherapy was effective and well tolerated and warrants further study in humans.

### 4.3. Bcl-6-Targeting PROTACs

The B-cell lymphoma 6 (Bcl6) molecule, which acts as a mandatory repressor for the proliferation of mature B lymphocytes, and immunoglobulin somatic hypermutation (SH) and class switching during the germinal center (GC) reaction, was identified as a target for PROTACs [86]. It halts the premature activation and differentiation of B lymphocytes, and allows the necessary DNA breaks required for the SH reaction that leads to the production of high-affinity antibodies, by repressing the expression of *ATR*, *CHEK1*, *TP53*, *CDKN1A*, *CDKN2A*, and *p14ARF* [87,88]. BCL6 is a proto-oncogene involved in the pathogenesis of DLBCL, FL, MCL, LBL, and BL [88,89,90,91]. In a study by McCoull et al., novel Bcl6-targeting PROTACs were designed and tested in a subset of DLBCL cell lines [92]. High permeability was one of the main requirements for the Blc6 ligands, in addition to potency and selectivity. A 5-chloro-pyrimidine was chosen over pyrazolo-pyrimidine, and further modifications were applied to enhance binding without increased kinase inhibitory activity. Thalidomide was used as an E3 ligand to mediate Bcl6 degradation through CRBN, and so PROTAC 15 was designed. Surprisingly, Bcl6 degradation with PROTAC 15 in an OCI-Ly1 cell line was dose-dependent but not complete, with an 82% effect reached at 1 µM. The degradation was evident after 1 h of incubation and remained detectable after 72 h, paralleled by an increase in Bcl6 levels. A similar level of Bcl6 degradation, ranging from 59% to 84%, was detected in DLBCL and Burkitt’s lymphoma cell lines. The next step was to evaluate the antiproliferative effect of PROTAC 15 on lymphoma cells, which was expected to exert a deeper phenotypic response through protein degradation rather than simple inhibition. However, PROTAC 15 failed to decrease the proliferative potential of the studied lymphoma cell lines (GCB and ABC DLBLC, and MM) more than a sheer inhibitor. Prolonged incubation of up to 16 days did not indicate that PROTAC 15 had a Bcl6-selective effect: Bcl6 could still be detected in all cellular compartments (nuclear, cytoplasmic, chromatin-bound fractions) despite treatment with PROTACs. The lack of complete degradation was consistent with similar PROTACs with different linkers; this finding could not be explained by less accessible fractions of Bcl2, as all compartments had proportionately reduced levels of Bcl6. The authors concluded that a combination therapy aiming at several distinct molecules that drive lymphoma growth may be sufficient to overcome its proliferative properties.

A different approach to Bcl6 inhibition was achieved by Kerres et al., who developed a Bcl6-targeting degrader that worked through the UPS [93]. The authors performed structural studies (broad complex, Tramtrack, Bric-a-Brac) to find an appropriate Bcl6 BTB domain-binding compound BI-3802 that degraded Bcl6 in human GCB-type DLBCL lymphoma cell lines at nanomolar concentrations. Moreover, the compound-induced Bcl6 depletion was strongly related to the presence of the zinc finger (ZF) DNA-binding domain of Bcl6, and it was concluded that its binding to DNA is indispensable. Degradation of Bcl6 rather than its inhibition led to a stronger gene derepression and upregulation. This effect might be explained by the presence of other regions outside the BTB for the transcriptional repression of Bcl6. The direct pathway leading to Bcl6 degradation via proteasome remained unknown, but it was demonstrated that the cullin-dependent protein ubiquitylation was not responsible for degradation in this case. Antiproliferative effects or growth reduction were seen in several DLBCL cell lines, but not in Toledo or MV-4-11, neither of which express Bcl6. However, the bioavailability of BI-3802 was poor and thus no animal xenograft studies were performed.

A study by Bellenie et al. employed optimized benzimidazolone degraders to target Bcl6 B-cell lymphoma via proteasome [94]. After a series of syntheses and structural studies, several promising compounds were developed. Incubation of GCB-type DLBCL cell lines SU-DHL-4 and OCI-Ly1 with molecule CCT368682 led to significant concentration-dependent Bcl6 degradation at concentrations starting from 0.156 µM. Moreover, the DC_50_ values were similar to or below the IC_50_ values, suggesting that efficient degradation may not require lengthy exposure. A more effective compound 1 (CCT369260) was developed and demonstrated full degradation of Bcl6 in OCI-Ly1 and SU-DHL-4 cell lines, at even smaller concentrations than CCT368682 (0.035 vs. 0.28 µM, and 0.092 vs. 0.35 µM, respectively). The degrading activity was accompanied by antiproliferative activity in both compounds, with a stronger effect on Bcl6 mediated by its depletion rather than its inhibition. Compound 1 was chosen for in vivo studies in an OCI-Ly1 xenograft model in Balb/C mice; results showed oral bioavailability of 54%. Free plasma concentrations of >1 nM led to significant degradation of Bcl6 in plasma and tumor samples, with a maximal effect seen at 4 h after administration. At 10 h the compound concentration dropped, and the Bcl6 levels rebounded at about 12 to 16 h after administration. Although not a typical PROTAC, compound 1 proved its efficacy and availability with oral administration. There was no “hook” effect during the experiments, probably due to the lack of bivalent binding, and competition between the dimers and the PROTAC-bound ternary complexes.

A distinct approach to Bcl6 degradation was demonstrated by Słabicki et al. [95], who designed a novel Bcl6-targeting molecule BI-3802 based on in silico studies, which induced rapid Bcl6 degradation at IC_50_ of less than 3 nM in the SU-DHL4 DLBCL cell line by binding the BTB domain. The degradation was found to be mediated via the proteasome through the non-cullin SIAH1 E3 ligase, but not the Cullin-RING E3 ligase. The process depended on the first 275 amino acids from the N-terminus and did not require the Zinc-finger domain. BI-3802 led to a formation of Bcl6 polymers of sinusoidal shape, present at distinct foci in the cells and demonstrating enhanced interaction with SIAH1. However, the mutations in the BTB domain prevented BI-3802 binding, and conferred resistance in Bcl6-dependent DLBCL cell lines.

### 4.4. BET—Targeting PROTACs

Mantle cell lymphoma relies on deregulated signaling and the activity of transcription factors, such as MYC and NFκB [96]. The bromodomain and extra-terminal domain (BET) family proteins are important transcriptional regulators in MCL [97]. They regulate gene transcription by binding acetylated lysine residues in histones [98]. The BET inhibitors induce apoptosis of primary MCL cells that are either sensitive or resistant to ibrutinib, coupled with a reduced expression of MYC, BCL2, CDK4/6, and NFκB targeted genes [99]. However, the inhibition is incomplete and reversible. BET PROTACs ARV-825 and ARV-771 consist of a BETi (OTX015/JQ1), a linker, and an E3 ligase ligand (pomalidomide for CRBN or a ligand for VHL, respectively) [100]. The treatment of MCL cell lines with ARV-771 and ARV-825 resulted in significant depletion of BRD4 and BRD2, in contrast to sole inhibition with OTX015, which led to an accumulation of BRD4 protein. Moreover, BET PROTACs induced apoptosis at nanomolar concentrations regardless of the MCL cell-line sensitivity to ibrutinib (Mino, Z138, MAVER-1, Granta-519). The IC_50_ values of induction of apoptosis in ibrutinib-sensitive MCL cells were around 17 nM for PROTACs, compared to 398 nM for OTX015 BETi. Z138 cells were resistant to BETi, but underwent apoptosis induced by PROTACs at IC_50_ values of 142 nM for ARV-771 and 327 nM for ARV-825. The primary MCL cells were less prone to PROTACs, and significant apoptosis was detected from about 50 nM. The addition of pomalidomide to BET PROTACs had little effect on MCL cells subjected to ARV-825. BET PROTACs exerted much less activity in normal B lymphocytes and CD34+ cord-blood-derived progenitor cells. The latter finding might be explained by the greater dependency of MCL cells on the oncoproteins driven by BRD4. The analyses of the mRNA transcript levels revealed that PET PROTACs reduced the mRNA levels for c-Myc, Bcl-2, PRDM1, cyclin D1, Bcl-xL, XIAP, c-FLIP, c-IAP2, Il10, and BTK. The protein levels in MCL cell lines were also decreased by BET PROTACs, including c-Myc, CDK4, cyclin D1, XIAP, MCL1, and Bcl-Cl. Similar observations were made upon treatment of patient-derived MCL cells. Xenograft mouse models showed greater potency of ARV-771, compared to BETi OTX015, for reducing MCL tumor growth in vivo and improving survival. The BET PROTAC was well tolerated with no significant weight loss in mice after the successful treatment. The combinations of ARV-771 with ibrutinib, venetoclax, or palbociclib demonstrated synergistic effects on ibrutinib-sensitive and ibrutinib-resistant MCL cells in vitro. 

The bromodomain and extra terminal (BET) family of proteins regulates gene transcription by binding acetylated lysine residues in histones [98]. One of its proteins, BRD4, is widely expressed in mammalian cells and is indispensable during mitosis. BRD4 was found to be necessary for AML growth, and its inhibition with small-molecule compound JQ1 led to the elimination of leukemic cells in vitro and in vivo [101]. Its clinical effect was accompanied by transcriptional repression of c-Myc, p-STAT5, PIM1, and CDK4/6. However, continuous treatment with BETi (i.e., OTX015) was found to lead to the rebound accumulation of BRD4, and loss of antileukemic effect. In response, BET-PROTAC ARV-825 was designed with the BRD4-binding moiety (OTX015) and pomalidomide as its CRBN-recruiting part [102]. BET-PROTAC led to a prolonged and profound decrease of BET proteins (including BRD4) in post-myeloproliferative (post-MPN) AML cell lines and patient-derived post-MPN AML cells, compared to BETi at equimolar concentrations. A synergistic effect was observed upon co-incubation with ruxolitinib, leading to significant depletion of BRD4, c-Myc, p-STAT5, Bcl-xL, PIM1, and CDK4/6. A more bioavailable BET-PROTAC ARV-771, which recruits VHL E3 ligase, was also more potent in the murine xenograft model of secondary AML, compared with BET inhibitors, in terms of the reduction of tumor growth and overall survival, without significant impact on the animals’ weight [102].

### 4.5. BCR-ABL1—Targeting PROTACs

Chronic myelogenous leukemia is a myeloproliferative neoplasm that is driven by a *BCR-Abl* fusion gene, resulting in constitutive activation of a tyrosine-protein kinase BCR-ABL1 [103]. The treatment involves oral tyrosine kinase inhibitors (TKI) and there are currently several active compounds that allow long-lasting responses. However, point mutations occur in the ATP-binding site, and about 15–20% of CML patients receiving imatinib fail to achieve responses. Subsequent generations of TKIs have helped in resistant cases, but mutations continue to occur against the newest TKIs with poor prognosis for patients. One approach is to reduce the expression of the BCR-ABL protein. Omacetaxine suppresses the ribosome and inhibits its synthesis [104], but the drug is not well tolerated in patients with CML, probably due to the inhibition of the total protein synthesis, and a high percentage of these patients withdraw from therapy. Another method of reducing BCR-Abl kinase activity is to reduce the concentration of the existing protein by degradation. Targeted BCR-Abl degradation may be mediated by SNIPERs or PROTACs.

The activity of PROTACs is not restricted to the binding site within a kinase domain of the POI and might also affect the allosteric sites. Such an idea was applied in a study using an allosteric ABL1 inhibitor GNF-5 on CML cell lines and primary CML samples [105]. GNF-5 was chosen from several GNF compounds and found to link to the VHL ligand. Further improvements in activity and cell permeability led to the discovery of GMB-475. The PROTAC induced the degradation of BCR-ABL1 and c-ABL1 in a human CML cell line and murine BCR-ABL1 transformed cells, and inhibited cell proliferation with an IC_50_ of approximately 1 µM. Combination treatment with imatinib resulted in a synergistic effect that lowered the IC_50_ for imatinib threefold. Little additional effect was seen for ponatinib. Furthermore, GMB-475 exerted a more potent effect downstream of the BCR-ABL1 (via pERK and pCRKL pathways), compared to the sole inhibitor. The CML progenitor cells showed marked apoptosis after incubation with GMB-475, and stem cells presented only a minor increase in the apoptosis rate, which confirmed their independence from BCR-ABL1 signaling. Moreover, these stem cells were found to confer CML survival and progression.

### 4.6. BRAF-V600E—Targeting PROTACs

Another protein of interest is a mutated BRAF-V600E kinase with a Ras-RAF-MEK-ERK pathway, which plays an important role in the pathogenesis of several human malignancies including hairy cell leukemia [106]. Three BRAF kinase inhibitors, vemurafenib, dabrafenib, and encorafenib, have been used in the treatment of patients with melanoma and hairy cell leukemia. However, drug resistance occurs with the loss of a binding spot on BRAF, or kinase inhibitors become inactive for alternative classes of BRAF mutants [107]. A research team designed an SJF-0628 PROTAC based on vemurafenib, recruiting VHL E3 ligase [108]; the compound potently degraded BRAF-V600E (DC_50_ of 6.8 nM), as well as a splice variant BRAF-p61V600E (DC_50_ of 72 nM), BRAFG469A (DC_50_ of 15 nM), and a kinase-dead/hypoactive BRAFG466V variant (DC_50_ of 23 nM), in multiple cell lines. The wild type of BRAF was spared, however, in cell lines with an activated conformation of BRAF through either the amplified receptor tyrosine kinases or mutant RAS (constitutive upstream signaling). Pretreatment with appropriate inhibitors desensitized BRAF^WT^ to the studied PROTAC. On the other hand, inhibition of downstream signaling by trametinib or cobimetinib (MEK inhibitors) resulted in an increased BRAF^WT^ ubiquitination. SJF-0628 PROTAC was efficient in inducing a decrease in cell growth in several cell lines harboring mutated BRAF and was successful at reducing the size of tumors in murine xenograft models of melanoma. In a study by Han et al., vemurafenib and BI882370 (a pan-RAF inhibitor) were linked to thalidomide [106]. Reductions of BRAFV600E protein were observed in a melanoma cell line from 12 nM with both of these compounds when used as warheads in PROTACs, but were not detected in BRAF^WT^-harboring lung cancer cells. Both PROTACs impaired cell growth of melanoma and colon cancer cells in vitro, but not of non-neoplastic cells.

### 4.7. STAT3-Targeting PROTACs

Signal transducer and activator of transcription 3 (STAT3) is a transcription factor that belongs to the JAK-STAT signaling pathway, and becomes activated in response to cytokines, growth factors, MAPK, and c-src non-receptor tyrosine kinases [109]. Overexpression or increased activation of STAT3 is implicated in several hematologic malignancies, including acute leukemias, chronic lymphocytic leukemia, chronic myelogenous leukemia, large granular leukemia, as well as Hodgkin’s and non-Hodgkin’s lymphomas, and multiple myeloma [110]. Bai et al. designed a cell-permeable PROTAC SD-36 that selectively targets STAT3 and utilizes lenalidomide for a CRBN ligand [111]. SD-36 potently degraded STAT3 in a concentration- and time-dependent manner in MOLM-16 (AML), and SU-DHL-1, DEL, and KI-JK (ALCL) cell lines at 250 nM. SD-36 displayed selectivity for STAT3 rather than other STAT members (Kd values of 50 nM compared with 1–2 µM and over 10 µM). Moreover, SD-36 mediated degradation of D661Y, K658R, and Y705F mutant STAT3 proteins in cells. Expression of many cancer-related genes was altered by SD-36 in MOLM-16 cells, i.e., down-regulated *BCL3*, *HCK*, *HGF*, *JAK3*, *PIM1*, *SOCS3*, and *VEGFA*. It was found that only one of nine AML cell lines demonstrated significant growth inhibition with SD-36, compared to five of nine lymphoma cell lines (only ALCL lymphoma lines). Genes that were downregulated by SD-36 in AML cells included BCL3 and JAK3, along with c-Myc protein in the ALCL cell line. SD-36 exerted a profound and prolonged xenograft tumor suppression even after a single dose of 25 mg/kg. The treatment was well tolerated with no significant toxicities and a transient weight gain.

### 4.8. HPK1—Targeting PROTACs

A different approach was taken by Si et al., using a PROTAC directed against the hematopoietic progenitor kinase 1 (HPK1), which mediates T cell dysfunction reflected by T cell exhaustion, diminished immunity against cancer cells, and shorter patient survival [112]. Continuous antigen exposure in the tumor microenvironment, and PD-1/PD-L1 interactions between tumor cells and T lymphocytes, render them inactive against or tolerant of the tumor [113]. Therapies based on disrupting such interactions (namely anti-PD-1 and anti-CTLA-4) have proved efficient in multiple human malignancies, including melanoma and Hodgkin’s lymphoma [114,115]. Si et al. developed SS44 and SS47 PROTACs, successfully targeting HPK1 and CRBN E3 ligase [112]. HPK1 degradation resulted in an increased proliferation of CD4+ and CD8+ T lymphocytes, and increased IFNγ production. Moreover, tumor growth in a murine breast cancer model was strongly inhibited by the PROTAC, and a synergistic effect was observed upon administration of an anti-PD-1 antibody. The degradation of HPK1 also augmented the in vivo antitumor reaction of the BCMA CAR-T cells against patient-derived multiple myeloma xenograft. The latter finding represents a novel modality (PROTACs) in cancer immunotherapy, by fine-tuning the CAR-T cell technology.

Most PROTACs rely on CRBN or VHL for E3 ligase recruiting. However, drug resistance might hinder the potential advantage of using PROTACs, as gene alterations appear in the core components of E3 ligase complexes and impede ligation with a ligand [116]. Such changes can develop under chronic treatment, resulting in the incomplete elimination of targeted tumor cells. However, the duration of the linkage between the E3 ligase and the POI may not be long enough to exert adequate ubiquitination [117]. There is a need for the discovery of new PROTAC E3 ligases with a specific expression profile, based on targeted protein expression depending on the tumor type, cellular compartment, or current cell condition. Moreover, newer generations of PROTACs might even be activated by light (photoPROTACs, PHOTACs) [118]. Other PROTACs act on the activation of the tyrosine kinase receptors and their subsequent phosphorylation (phosphoPROTACs) [119]. They sense the current state of the cell, to induce conditional protein degradation with phosphotyrosine-binding (PTB) or Src homology 2 (SH2) domains when these are most required. Another interesting modification of PROTACs is the development of HaloPROTACs [120], which include the VHL ligand and are capable of degrading HaloTag7 fusion proteins of interest at nanomolar concentrations. Their advantage over classical PROTACs results from their lower molecular weight, with a short motif including a chloroalkane with binds HaloTag7 fusion protein and a hydroxyproline derivative that binds VHL. This technology might improve studies on tagged or targeted protein degradation. Furthermore, homoPROTACs have been developed to induce self-degradation of the recruited E3 ligase by dimerization, i.e., VHL E3 ligase [121].

### 4.9. SNIPERs

Targeted protein degradation is also mediated by a group of protein-degrading chimeric small molecules, similar to PROTACs, called specific and non-genetic inhibitors of apoptosis protein (IAP)-dependent protein erasers (SNIPERs). SNIPERs hijack IAP ubiquitin ligases and lead to the degradation of the POI, as well as IAPs such as cIAP1 and XIAP [122]. This dual action seems to be ideally suited to addressing malignant cells, including hematologic malignancies (AML, ALL), that usually overexpress IAPs [123]. IAPs bind to caspases and Smac to inhibit apoptosis. Moreover, their RING domains have E3 ligase activity that mediates ubiquitination and proteasomal degradation of several substrates, including caspases, Smac, and IAPs themselves. IAPs induce NFκB activity through nondegradative polyubiquitination of inhibitory factor RIP1. Targeting IAPs showed promising results in several hematologic malignancies, from acute and chronic leukemias, to lymphomas and multiple myeloma (for review see [123]). The first generation of SNIPERs incorporated bestatin, and further studies revealed that substitution of bestatin with MV-1 and LCL as IAPs ligands improved their affinity. Combinations of new IAP ligands with several molecules can target specific proteins, i.e., imatinib, dasatinib, HG-7-85-01 to target Abl kinase in BCR-ABL-dependent diseases, or 4-OHT to target estrogen receptor in breast cancer [124,125].

In a study by Zhang et al., ABT-263 (navitoclax), which targets Bcl-xL with higher affinity than Bcl-2 or Bcl-w, was linked to LCL161 derivatives that potently recruited XIAP [126]. Several compounds with slight linker modifications were tested in vitro on MyLa 1929 (cutaneous T-cell lymphoma), MOLT-4 (T-ALL), and RS4;11 (B-ALL) cell lines. Compound 8a demonstrated the most significant activity in the abovementioned cell lines at a concentration of 500 nM after 8 h of incubation, with complete protein degradation after 16 h. Similar results were obtained in solid cancer cell lines (melanoma, breast cancer, colorectal cancer, non-small cell lung cancer). However, it was observed that cells with lower levels of XIAP, i.e., the triple-negative breast cancer cell line MDA-MB-231, were less sensitive to SNIPER-induced Bcl-xL degradation, compared with cells expressing higher levels of XIAP, i.e., non-small cell lung cancer cell line A549. Tailoring the SNIPER design to the treated disease may possibly overcome these insensitivity issues. The studied SNIPER 8a platelet toxicity appeared low compared with the toxicity of ABT-263, because of the low expression of IAPs in human platelets.

Shibata et al. designed a series of SNIPERs targeting BCR-Abl [124,127] and suggested that the shorter duration of the protein knockdown induced by SNIPERs is a disadvantage compared to PROTACs. They explained that SNIPERs offer dual action by degrading the POI, and that IAPs are implicated in promoting tumor development and resistance to anticancer drugs. After laborious experiments SNIPER(ABL)-39 was developed, which linked dasatinib with an LCL161 derivative. The compound was active at a concentration of 10 nM in reducing the BCR-Abl protein concentration, as well as suppressing the growth of CML cells in vitro even after the drug was removed from the culture. There is a requirement for new high-affinity IAP inhibitors that work at nanomolar concentrations in vivo.

## 5. Chaperone-Mediated Autophagy 

Chaperone-mediated autophagy (CMA) is a type of selective autophagy that degrades cytosolic proteins carrying a KFERQ-like motif recognized by cellular chaperones such as Heat Shock Cognate 70 (HSC70) protein [128]. HSC70 brings the targeted proteins to the lysosome membrane, where they become bound by the lysosome-associated membrane protein type 2A (LAMP2A) and unfold to cross the lysosomal membrane. The LAMP2A polymerizes and forms a CMA translocation complex. CMA and other means of cellular proteostasis work together, and inhibiting one leads to an upregulation of the others. However, incessant inhibition will eventually lead to the accumulation of protein aggregates, as observed in some degenerative diseases with impaired proteostasis [129]. The role of CMA in cancer is yet unknown and might depend on the type of malignancy and the stage of its development [130].

One of the earliest papers on CMA in hematologic malignancies was a study by Xia et al. [131]. They studied the effect of manipulating cellular metabolism to induce cancer cell apoptosis by the activation of protein degradation through CMA. Quizartinib (AC220), an FLT3 inhibitor, together with autophagy inhibitors A70 or C43, reduced cell viability of FLT3/ITD-mutated (Molm-14) and FLT3-unmutated AML cell lines (HEL, OCI-AML3). The combined treatment of solid cancer cell lines induced significant degradation of mutant p53, which carries a CMA-targeting motif, with no change in p53 mRNA levels. Moreover, HK2, an oncogenic kinase essential to preventing metabolic stress induced by glucose deprivation, was also degraded which led to the metabolic malfunction of cancer cells and a fatal drop in ATP levels. CMA activity and especially LAMP2 levels have been observed to correlate with responses to azacitidine (AZA) in MDS and AML cells in vitro [132]. All 3 isoforms of LAMP2 were reduced in AZA-resistant cells, possibly due to demethylation of the promoter of a *LAMP2* X-linked repressor. Depletion of LAMP2 resulted in inhibition of CMA and increased levels of MLLT11/AF1Q and BCL2L10, which mediate MDS cells’ resistance to AZA and their survival [133]. Low initial expression of LAMP2 was a predictor of refractoriness in patients with MDS or AML, and decreasing levels of LAMP2 during AZA treatment led to the development of resistance and a compensatory increase in autophagy. Inhibiting autophagy with a lysosome inhibitor, hydroxychloroquine, restored CMA function and allowed subsequent degradation of CMA substrates MLLT11/AF1Q and BCL2L10, which are also responsible for promoting leukemogenesis.

Facilitated CMA degradation was presented in a paper by Li et al., describing induced depletion of AF1q, a mixed-lineage leukemia fusion partner protein, in leukemic cells treated with CMA-inducing 6-aminonicotinamide [134]. However, AF1q was not modified or targeted for degradation, as it naturally contains a KFERQ-like motif which makes it amenable for lysosomal breakdown. The process was inhibited by chloroquine knockdown of LAMP2A or HSPA8, suggesting a CMA mechanism for AF1q degradation. An interesting observation was made by Vakifahmetoglu-Norberg et al. who observed enhanced mutant p53 degradation in dormant cancer (ovarian, breast, fibrosarcoma) cells under nonproliferating conditions (confluent growth) [135]. The effect was induced by the suppression of macroautophagy with a small molecule spautin-1, mediated by CMA, and led to apoptosis in vitro. Perhaps similar observations might be made in relation to several hematologic malignancies that rely on mutated p53, such as CLL.

## 6. Lysosome Modification

PROTAC technology seems to be a promising strategy, but there are several limitations to its usage: high molecular weight and therefore reduced bioavailability; cell permeability or blood-brain barrier penetration; dependence on certain ligases within specific cell types; proteasome resistance of distinct proteins or non-protein molecules; development of mutations resistant to PROTACs; and lack of effect on proteins without ligandable cytosolic domains. Lysosomal degradation might become an alternative and occurs independent of the proteasomal pathway [136] (Figure 2). The lysosomal pathway involves a family of cell-membrane lysosomal targeting receptors (LTRs) that capture extracellular or membrane proteins which then become engulfed and fuse with the lysosome for subsequent degradation [137].

One of the first utilized LTRs was the cation-independent mannose-6-phosphate receptor (CI-M6PR) or insulin-like growth factor 2 receptor (IGF2R), which binds proteins with N-glycans capped with mannose-6-phosphate (M6P) and is widely expressed among different cell types [138]. An initial lysosome targeting chimera (LYTAC) was created by binding a polyclonal anti-mouse IgG antibody to M6P glycopolypeptides. Preliminary experiments with biotinylated polyM6P revealed increased and continuous uptake of avidin in numerous human cell lines (AML, ALL, CML, Hodgkin lymphoma, as well as breast cancer, and cervical cancer). The receptor recycling allowed for an ongoing accumulation of fluorescence in the cells in the endosomes and lysosomes. An experiment using LYTAC with apolipoprotein E4, responsible for neurodegenerative diseases, showed a sustained 13-fold increase in absorption. LYTAC was also effective in the degradation of a membrane-bound extracellular protein EGFR, extending up to 72 h after the initial incubation. Moreover, an anti-PD-L1-M6Pn LYTAC was tested in MDA-MB-231 breast cancer cells and HDLM-2 Hodgkin lymphoma cells [138]. The treatment resulted in an average 33% reduction in cell-surface expression of PD-L1 in breast cancer cells at 72 h, and an average 50% decrease after 36 h in Hodgkin lymphoma cells. For an indication of the AUTAC and LYTAC strategies in development, see Table 2.

## 7. Other Modalities in Targeted Protein Degradation

Protein degradation can be prompted by an unfolded protein response via the UPS. One of the reasons might be the hydrophobicity of the protein surface, and hydrophobic tagging of proteins is another strategy used for targeted protein degradation [118]. Ma et al. used this approach to target and deplete EZH2 histone methyltransferase [143]. The over-function of EZH2 is implicated in the pathogenesis of several hematologic malignancies, such as B-cell and T-cell lymphomas, myeloblastic syndrome (MDS), and acute and chronic leukemias [144,145,146,147,148]. Some EZH2 inhibitors have already entered clinical trials. In a recent study, a non-covalent EZH2 inhibitor was linked to a bulky hydrophobic adamantyl group to form a first-in-class EZH2 selective degrader MS1943 [143]. The compound was active in reducing the targeted protein in triple-negative breast cancer, as well as KARPAS-422 and SUDHL8 lymphoma cell lines, at a concentration of 4 µM (IC_50_ 120 nM). MS1943 also demonstrated significant selectivity (<10% of inhibition) at 10 µM against various methyltransferases, kinases, protein-coupled receptors, ion channels, and transporters. The compound induced prolonged endoplasmic reticulum stress in sensitive but not refractory breast cancer cells, as reflected by the increased expression of the unfolded protein response genes *Xbp1*, *Chop*, and *Bip*. A murine xenograft model of triple-negative breast cancer proved the efficacy of MS1943, with complete tumor growth suppression at daily dosing for seven days and good overall tolerance. Based on those results, the authors concluded that MS1943 could be effective in EZH2-dependent malignancies.

## 8. Conclusions

Targeted treatment modalities have dramatically changed the therapeutic options and outcomes for patients with hematological malignancies. One of the first small molecules aimed at disease pathogenesis was imatinib, a tyrosine kinase inhibitor for *BCR-ABL*-driven chronic myelogenous leukemia. However, resistance to imatinib eventually develops, leading to a lack of direct contact with the kinase due to mutations occurring at the binding site. A more recent idea has been to degrade a protein of interest (POI) rather than inhibit its activity, as some essential proteins are simply non-enzymatic. Cellular protein degradation systems can potentially be harnessed for targeted degradation of POIs. It has been shown that such an approach allows the effective depletion of POIs, including non-enzymatic targets previously deemed non-druggable. Several concerns have already emerged in relation to such therapies, including the appropriate tagging of POIs to be detected by the corresponding degrader, choice of an optimal tissue-specific degradation enzyme, and overcoming the compensatory activation of alternative signaling pathways, among other issues. Perhaps the future lies in the use of combination therapies at various key points in the pathomechanism of a given disease. Moreover, such treatment should be relatively non-toxic and preferably administered orally, to ensure good quality of life and enable home treatment. The latter turned out to be extremely important during the COVID-19 pandemic, when it was necessary to maintain social distance, especially for frail people with malignancies. Research into targeted protein degradation will continue to develop, and promising in vitro and in vivo results warrant its further study.

## Figures and Tables

**Figure 1 cancers-14-03778-f001:**
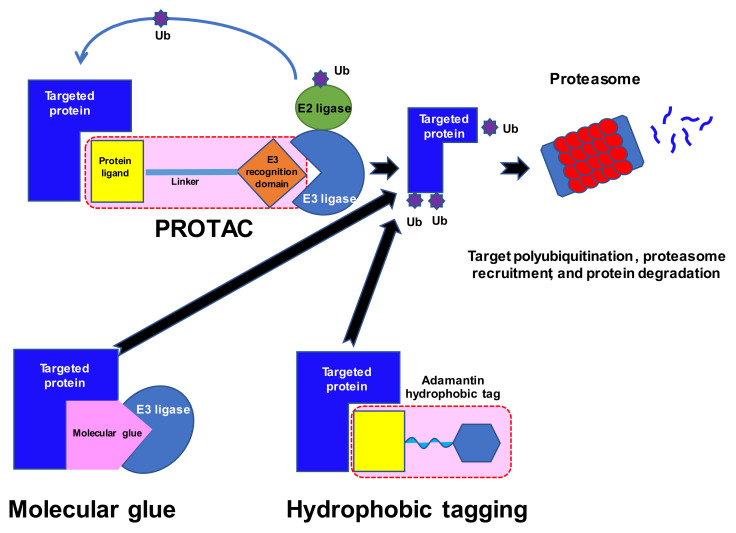
Ubiquitin–proteasome system for cellular protein degradation. Ub, ubiquitin; E3, E3 ubiquitin ligases; PROTAC, proteolysis-targeting chimera.

**Figure 2 cancers-14-03778-f002:**
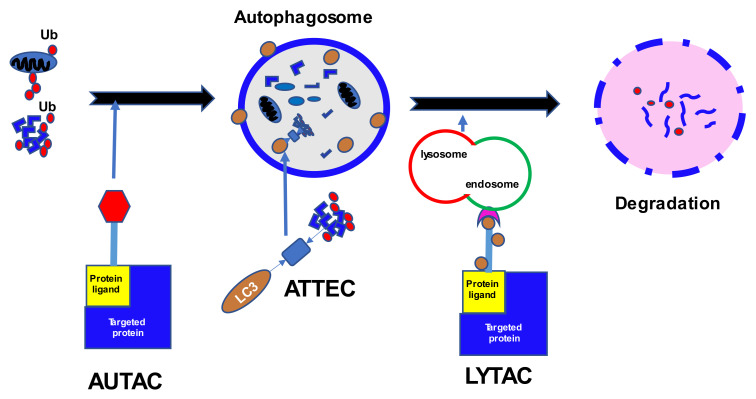
Cellular protein degradation through autophagy. Ub, ubiquitin; LC3, microtubule-associated proteins; AUTAC, autophagy-targeting chimeras; ATTEC, autophagosome-tethering compound; LYTAC, lysosome-targeting chimeras; E3, E3 ubiquitin ligases; PROTAC, proteolysis targeting chimera.

**Table 1 cancers-14-03778-t001:** The currently studied PROTACs in clinical trials.

PROTAC	E3 Ligase Targeting Ligand	E3 Ligase	Target	Disease	Phase	No. of Patients	NCT
ARV-110	Lenalidomide-based	Cereblon (CRBN)	Androgen receptor	Metastatic castration-resistant prostate cancer	1	40	NCT05177042
ARV-110	thalidomide-based	Cereblon (CRBN)	Androgen receptor	Metastatic castration-resistant prostate cancer	1/2	250	NCT03888612
ARV-766	undisclosed	undisclosed	Androgen receptor	Metastatic castration-resistant prostate cancer	1	60	NCT05067140
CC-94676	undisclosed	Cereblon (CRBN)	Androgen receptor	Metastatic, castration-resistant prostate cancer	1	70	NCT04428788
ARV-471	Lenalidomide-based	Cereblon (CRBN)	Estrogen receptor	ER+/HER- advanced or metastatic breast cancer	1/2	170	NCT04072952
DT2216	undisclosed	Von Hippel-Lindau (VHL)	BCL-xL	Solid and hematological malignancies	1	24	NCT04886622
NX-2127	IMID-based	Cereblon (CRBN)	BTK	Hematological malignancies (BTK C481-mutated CLL/SLL, WM, MZL, FL, DLBCL)	1	130	NCT04830137
DKY709	Pomalidomide-based	Cereblon (CRBN)	IKZF2	Solid malignancies	1	380	NCT03891953

**Table 2 cancers-14-03778-t002:** LYTAC and AUTAC strategies in development.

**LYTAC**				
	**Lysosome Shuttling Receptor and Ligand**	**POI Ligand**	**POI**	**Ref.**
1st generation	CI-M6PR, N-carboxyanhydride-derived glycopeptide	mAb: cetuximab,	apolipoprotein E4 (ApoE4), epidermal growth factor (EGFR), CD71 and PD-L1	[138]
2nd generation—tissue specific	liver-specific asialoglycoprotein receptor (ASGPR), N- acetylgalactosamine (tri-GalNAc)	mAb: cetuximab, pertuzumab; polyspecific integrin-binding peptide (PIP),	EGFR, HER2, integrins	[139]
		Small molecule	Human macrophage migration inhibitory factor (MIF)	[140]
**AUTAC**				
**Autophagosome receptor and ligand**		**POI ligand**	**POI**	**Ref.**
LC3, p62		Synthetic PHTPP, vinclozolin, fumagillin, 4-phenylbutyric acid (PBA), Anle138b	estrogen receptor beta (ERβ), androgen receptor, MetAP2, misfolded proteins i.e., tau	[141]
		Fumagillin, synthetic ligand of FKBP (SLF), JQ1 acid	MetAP2, FK506-binding protein, Brd4	[142]

## Data Availability

Pubmed scientific base.

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
