# Peer review of "Targeting Protein Degradation Pathways in Tumors: Focusing on their Role in Hematological Malignancies"

_cancers, 2022, doi:10.3390/cancers14153778_

Round 1

Reviewer 1 Report

This was an interesting, well written and timely review.  The authors do a thorough review of pathways of normal protein degradation and matching the basic science with translational efforts for developing PROTACS, AUTAC and Lytac strategies for targeting degradation of cancer related targets.

Minor concerns 

Simple summary line 9 "Amino Acid etc" needs more clarification

Line 140: Consider change from short protein 10 AA to peptide 

Line 235: I thought was confusing "induce autophagy of PML-RARapha or do authors mean compounds induce autophagy leading to degradation of  target protein?

Table 1 :  Adding the E3 ligase targeting ligand and E3 ligase may increase detail of the table for readers

A table of current stage of development of Lytac and AUTAC strategies even if all are preclinical may lead to a nice summarization of targets and chimeric strategies  for the reader

Author Response

Dear Reviewer,

thank you for the comments. Below please find our response to your comments/suggestions:

  1. Simple summary line 9 "Amino Acid etc" needs more clarification - Has been edited: 'etc' has been removed;
  2. Line 140: Consider change from short protein 10 AA to peptide - has been changed according to the Reviewer's suggestion;
  3. Line 235: I thought was confusing "induce autophagy of PML-RARapha or do authors mean compounds induce autophagy leading to degradation of  target protein? - that is correct - PML-RARalpha is degraded either by the UPS or by the lysosomal pathway - reference 46 and 47 are about these two ways;
  4. Table 1 :  Adding the E3 ligase targeting ligand and E3 ligase may increase detail of the table for readers - has been added although some information is still undisclosed by the researchers and the table may sound vague - especially the E3 ligase targeting ligand;
  5. A table of current stage of development of Lytac and AUTAC strategies even if all are preclinical may lead to a nice summarization of targets and chimeric strategies  for the reader - a table has been added.

Thank you for your review.

Kind regards,

Piotr Smolewski

Reviewer 2 Report

Overall this is a timely and comprhensive review of the expanding field of PROTACs in haematological malignancies, however, I believe it requires reformatting to be considered for publication. Comments are outlined below:

Since PROTACS harness the UPS the main focus when providing background should be on this with less emphasis and detail required on the UPR.

Page 2 line 54 - abnormal protein production is a consequence rather than a cause of multiple myeloma

Page 4 line 179 - define RP

Figure 1 and 2 - include a more detailed figure legend

Section 4 PROTACs and SNIPERs - this section requires reformatting

- please provide a clearer overview of PROTACs to include the small number of E3 ligases that are currently being used in PROTACs.

- please organise the text into subheadings for either PROTACs employing specific E3s or targeting specific proteins of interest. At present it is just a large block of text that is difficult to get through

Author Response

Dear Reviewer,

thank you for your comments and suggestions. Please find below the response to each point mentioned:

1. Since PROTACS harness the UPS the main focus when providing background should be on this with less emphasis and detail required on the UPR - the chapter on the UPR has been shortened to match the overall aim of the article.

2. Page 2 line 54 - abnormal protein production is a consequence rather than a cause of multiple myeloma - has been reworded;

3. Page 4 line 179 - define RP - has been defined;

4. Figure 1 and 2 - include a more detailed figure legend - has been added;

5. Section 4 PROTACs and SNIPERs - this section requires reformatting:

- please provide a clearer overview of PROTACs to include the small number of E3 ligases that are currently being used in PROTACs - a comment has been added.

  • please organise the text into subheadings for either PROTACs employing specific E3s or targeting specific proteins of interest. At present it is just a large block of text that is difficult to get through - subheadings for specific POIs have been added.

Kind regards,

Piotr Smolewski